

# Geoelectrical and hydro-chemical monitoring of karst formation at the laboratory scale

Flore Rembert[1], Marie Léger[2], Damien Jougnot[3], and Linda Luquot[2]

[1]Univ. Orléans, CNRS, BRGM, ISTO, UMR 7327, Orléans, F-45071, France
[2]Géosciences Montpellier, Univ Montpellier, CNRS, Univ Antilles, Montpellier, France
[3]Sorbonne Université, CNRS, EPHE, METIS, F-75005, Paris, France

**Correspondence:** Flore Rembert (flore.rembert@univ-orleans.fr)

**Abstract.** Ensuring sustainable strategies to manage water resources in karst reservoirs requires a better understanding of the mechanisms responsible for conduits formation in the rock mass and the development of detection methods for these hydrological and geochemical processes. In this study, we monitored the electrical conductivity of two limestone core samples during controlled dissolution experiments. We interpret the results with a physics-based model describing the porous medium

as effective structural parameters that are tortuosity and constrictivity. We obtain that constrictivity is more affected by calcite dissolution compared to tortuosity. Based on our experimental results and data sets from the literature, we show that the characteristic Johnson length is a valuable structural witness of calcite dissolution impact linking electrical and hydrological properties.

**Short summary.** The formation of underground cavities, called karsts, resulting from carbonate rock dissolution, is at stake in many environmental and societal issues, notably through risk management and the administration and quality of drinking water resources. Facing natural environment complexity, we propose a laboratory study combining hydro-chemical monitoring, 3D imaging, and non-invasive observation of electrical properties, showing the benefits of geoelectrical monitoring to map karst formation.





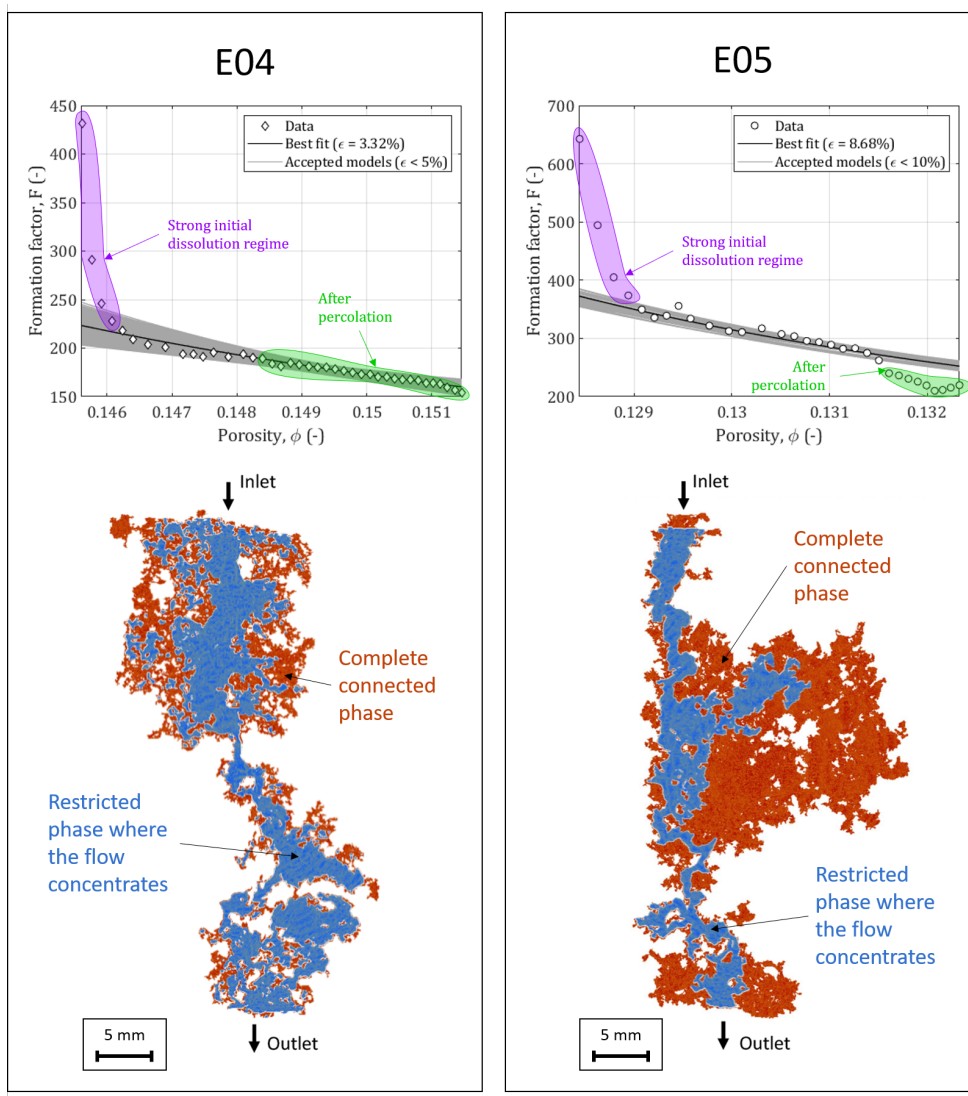

**Figure 0.** Graphical abstract

# 1 Introduction

Carbonate rocks occupy a vast area on the Earth's surface (Chen et al., 2017) and constitute reservoirs for key resources such as groundwater (e.g., Kačaroğlu, 1999; Bakalowicz, 2005), geothermal energy (e.g., Montanari et al., 2017), or fossil energies (e.g., Burchette, 2012). The study of carbonate rocks is also very active because of their potential to serve for carbon dioxide ($CO_2$) geologic sequestration (e.g., Luquot and Gouze, 2009; Cherubini et al., 2019). Carbonate rocks are also well known to be linked to coastal or agricultural issues with contamination, erosion, and landsliding, to civil engineering with risks of cavity



presence and collapsing, but also as a commonly exploited noble building material (e.g., Drew et al., 2017; Buckerfield et al., 2020). Investigating carbonate reservoirs is a crucial challenge due to the multi-scale heterogeneity of rock properties and their strong chemical reactivity (e.g., Choquette and Pray, 1970; Lønøy, 2006). These features are responsible for specific processes occurring in carbonate rocks over a wide size range (nm to km), such as groundwater flow and ionic transport in a reactive

porous medium (e.g., Ford and Williams, 2013).

Dissolution of carbonate samples caused by $CO_2$ or acid solution injection has already been well studied to understand the formations of wormholes and their complicate relationship with transport properties as permeability and porosity (e.g., Golfier et al., 2002; Noiriel et al., 2004, 2005; Rötting et al., 2015). However, these experiments were generally based on images analysis which is an accurate technique for laboratory works but cannot be used on the field. Additionally, in a subsurface

context, chemical analysis of the pore water can be quite intrusive, providing only restricted and spatially limited information (e.g., Goldscheider et al., 2008). Thus, studying large scale heterogeneities such as karst environments can benefit from the use of non-invasive tools such as the ones proposed in hydrogeophysics (e.g., Hubbard and Linde, 2011; Binley et al., 2015). In particular, geoelectrical methods are good candidates to detect the emergence of a sinkhole, to identify infiltration area, or to map ghost-rock features (e.g., Liñán Baena et al., 2009; Chalikakis et al., 2011; Meyerhoff et al., 2014; Kaufmann and

Romanov, 2016) since they present a high sensitivity to physical and chemical properties of both porous matrix and interstitial fluids (e.g., Glover, 2015). An increasing amount of work has shown the interest and the effectiveness of geoelectrical methods for laboratory or in situ monitoring of pore space description, hydrological processes, and contaminant transport (e.g., Revil et al., 2012; Garing et al., 2014; Jougnot et al., 2018; Ben Moshe et al., 2021; Sun et al., 2021; Koohbor et al., 2022). Studies on the geolectrical monitoring of calcite precipitation and dissolution processes are emerging, highlighting their interest for

non-intrusive characterizations (e.g., Wu et al., 2010; Zhang et al., 2012; Leroy et al., 2017; Cherubini et al., 2019; Niu and Zhang, 2019; Saneiyan et al., 2019, 2021; Izumoto et al., 2020, 2022; Rembert et al., 2022). Nevertheless, geophysical methods are indirect, and thus, require appropriate models to give a quantitative interpretation (e.g., Binley and Kemna, 2005; Kemna et al., 2012).

The present work attempts to give some answers on how the electrical signal is impacted by conduits formation in limestone

due to calcite dissolution, and how the electrical properties can be related to evolving structural parameters. To this end, acid injections are conducted on two core samples of the same limestone at atmospheric conditions and under two different flow rates. The electrical, chemical and hydrodynamic properties are recorded during the experiments, and the samples are characterized with laboratory and images methods before and after experiments. Evolution of all these properties coupled with the initial conditions, is analyzed in order to observe in which proportion they are responsible for the changes induced by the

acid flowing through the rock. Then, we focus on the electrical conductivity interpretation and apply a physics-based model that conceptualizes the porous medium as sub-parallel tortuous capillaries, which follow a pseudo-fractal size distribution and present sinusoidal variations of their aperture. Finally, we compare the electrical conductivity results of this study with experimental and numerical data sets from the literature.



## 2 Materials and Methods

### 2.1 Samples properties and petrophysical characterization


The two core samples of this study are part of a wider published data set (Leger and Luquot, 2021; Leger et al., 2022a). They were cored into an Oxfordian crinoid limestone from the Euville quarry in Nancy, France. The two samples are named E04 and E05. The cores have a length of 32 and 31 mm, respectively, and a diameter of 18 mm. They are surrounded with epoxy resin and PVC pipe for a total diameter of 25 mm (i.e., one inch).

X-ray Diffraction (XRD) on a Bruker D8 Discover, X-ray Fluorescence (XRF) measurements show that these rock samples are fully composed of calcium carbonate ($CaCO_3$). To characterize the cores samples' structure, petrophysical measurements are done by non-destructive laboratory methods: porosity $\phi$ (–), permeability $k$ ($m^2$), and electrical conductivity $\sigma$ (S $m^{-1}$) on saturated samples at different salinities to determine the formation factor $F$ (–) and the surface conductivity $\sigma_s$ (S $m^{-1}$), assuming

$$\sigma = \frac{\sigma_w}{F} + \sigma_s, \tag{1}$$

where $\sigma_w$ (S $m^{-1}$) is the saturating brine conductivity. Note that the surface conductivity can usually be neglected in carbonate material ((e.g., Cherubini et al., 2019)). The formation factor is an important parameter that can be related to microstructural properties through appropriate petrophysical relationships (e.g., Rembert et al., 2020). Among others, it can be related to the porosity of the medium through Archie (1942),

$$F = \phi^{-m_c}, \tag{2}$$

where $m_c$ (–) is the cementation exponent. Many studies use this relationship to find a unique value of $m_c$ to characterize their rock type. From this, for typical carbonate rocks the cementation exponent is comprised between 2 and 3 (e.g., Lucia, 1983). However, for highly heterogeneous rocks and samples submitted to dissolution, it has been found that Archie's equation is not suitable (e.g., Garing et al., 2014; Niu and Zhang, 2019; Revil et al., 2014). Nevertheless, it has been shown that the

formation factor is an interesting proxy to quantify the pore space geometrical properties, such as the tortuosity. Clennell (1997) proposes an extensive review of seminal works on tortuosity, among which the model of Winsauer et al. (1952) that describes the electrical tortuosity $\tau_e$ (–) as

$$\tau_e = \sqrt{F\phi}. \tag{3}$$

Clennell (1997) also discusses the relationship between electrical and hydraulic tortuosities, $\tau_h$ (see also discussions in Thanh
et al., 2019). Recent developments in petrophysics tend to develop physics-based models that rely on more accurate conceptualization of the porous medium geometries (e.g., Rembert et al., 2020; Soldi et al., 2017; Vinogradov et al., 2021).

In addition, X-Ray MicroTomographic (XRMT) images of the two core samples are collected with a pixel size of 12 μm and are analyzed with a homemade software to obtain complementary information about the samples structure. In particular, a pore size distribution is obtained from a probabilistic method displaying the chord lengths.



For further details about rock type, calculation methods and protocols, complete results, and discussion on the samples properties, the reader will find the information in Leger and Luquot (2021), Leger et al. (2022b), and Leger et al. (2022a). After dissolution, in order to estimate the impact of the experiments on the core properties, the same laboratory and imaging characterization as before the experiments are conducted on the samples.

## 2.2   Experimental setup and monitoring

The experimental protocol followed here is the same as the one used in Leger et al. (2022a) and Leger et al. (2022b). Under atmospheric pressure and temperature conditions, acidic solution is injected through the samples. The injected solution is composed of water previously balanced with samples of the same rock type mixed with acetic acid ($CH_3CO_2H$), and sodium acetate ($CH_3CO_2Na$). The resulting acid concentration is about $10^{-2}$ mol L$^{-1}$ buffered with a pH of 4. Figure 1 displays the homemade experimental device used for the percolation experiments.

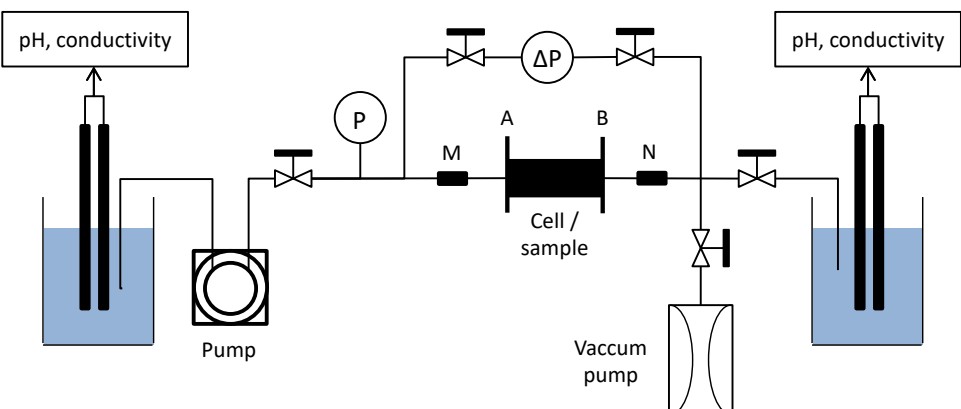

**Figure 1.** Experimental setup used for the percolation experiments. P is the absolute pressure sensor, $\Delta P$ is the differential pressure sensor. A, B, M and N are the electrodes, with A and B for the electric current injection and M and N for the electric potential measurement.

The inlet acid solution is stored in a beaker where electrical fluid conductivity and pH are monitored continuously, and is injected through the core sample by a peristaltic pump.

    Sensors of absolute and differential pressure continuously record the pressure difference between the sample inlet and outlet. It allows to calculate the evolution of the samples permeability $k$ (m$^2$) during the experiment using Darcy's law

$$k = \frac{Q\,L\,\mu_w}{A\,\Delta P},\tag{4}$$

where $Q$ (m$^3$ s$^{-1}$) is the flow rate, $L$ (m) is the sample length, $\mu_w = 0.001$ Pa s is the dynamic viscosity of water, $A$ (m$^2$) is the area of fluid injection, and $\Delta P$ (Pa) is the differential pressure between the inlet and outlet.



In order to evaluate the impact of the hydraulic conditions, E04 and E05 were flushed with two different flow rates, inducing two different transport conditions. These two experiment conditions are associated to two Péclet numbers $Pe$ (–) defined by

$$Pe = \frac{ul}{d}, \tag{5}$$

where $u$ (m s$^{-1}$) is the flow velocity injected in each sample, $d = 5\,10^{-9}$ m$^2$ s$^{-1}$ is the mean value of diffusion coefficients for ions presented in the fluid at $25\,°$C, and $l$ (m) is the characteristic length, which value is based on the pore size distribution calculated in Leger and Luquot (2021). The corresponding Péclet numbers are $Pe_{\mathrm{E04}} = 5.26$ and $Pe_{\mathrm{E05}} = 1.97$. They correspond to flow rates $Q_{\mathrm{E04}} = 6.0\,10^{-9}$m$^3$ s$^{-1}$and $Q_{\mathrm{E05}} = 2.5\,10^{-9}$m$^3$ s$^{-1}$.

The setup is also equipped with a 4-wire half bridge array to monitor the water-saturated sample electrical resistance $R$ ($\Omega$) with a CR1000 data logger from Campbell Scientific. The measured resistance is converted into the conductivity term $\sigma = 1/(R\,k_g)$, where $k_g$ (m) is the geometrical coefficient determined from the electrodes shape, size, and distance, using 3D numerical computing on the Electrical Impedance and Diffuse Optical tomography Reconstruction Software (EIDORS). We obtain $k_{g\,\mathrm{E04}} = 36$ mm and $k_{g\,\mathrm{E05}} = 37$ mm.

At the outlet, fluid samplings of 5 mL are continuously collected during day time and punctually during night time. pH, brine conductivity and cation concentrations are measured for each fluid sample. From calcium (Ca$^{2+}$) concentration monitoring, we can determine the mass of dissolved calcite through time, and thus, obtain the time variations of the rock sample porosity $\phi$ (–). At time $t_i$ (s) we have $\phi_{t_i} = \phi_{t_{i-1}} + \phi_{dt}$, where $\phi_{dt}$ (–) is the porosity difference over the time laps $dt = t_i - t_{i-1}$ (s). By definition, $\phi_{dt} = V_{dt}/V_{ech}$, where $V_{ech} = AL$ (m$^3$) is the core sample volume and $V_{dt}$ (m$^3$) is the pore volume difference over time laps $dt$. The latter is calculated as follows

$$V_{dt} = \frac{m_{\mathrm{CaCO_3}\,dt}}{\rho_{\mathrm{CaCO_3}}} = \frac{([\mathrm{Ca}^{2+}]_{t_i} - [\mathrm{Ca}^{2+}]_{t_0})\,dt\,Q\,M_{\mathrm{CaCO_3}}}{\rho_{\mathrm{CaCO_3}}}, \tag{6}$$

where $m_{\mathrm{CaCO_3}\,dt}$ (g) is the mass of calcium carbonate over time laps $dt$, $\rho_{\mathrm{CaCO_3}}$ (g m$^{-3}$) is the density of calcium carbonate, $[\mathrm{Ca}^{2+}]_{t_i}$ (mol m$^{-3}$) is the calcium concentration at the sampling time $t_i$, $[\mathrm{Ca}^{2+}]_{t_0}$ (mol m$^{-3}$) is the initial calcium concentration, and $M_{\mathrm{CaCO_3}}$ (g mol$^{-1}$) is the molecular masses of calcium carbonate.

## 2.3 Analytical model

### 2.3.1 Capillary Size Distribution

To relate the electrical monitoring with the porous medium microstructure, we use the Bundle Of Capillary Tubes (BOCT) approach (e.g., Vinogradov et al., 2021) It uses a bundle of parallel tortuous and constrictive capillaries as a conceptualization of the pore structure (e.g., Guarracino et al., 2014; Rembert et al., 2020). The conceptual geometry is represented in Fig. 2.

The BOCT model considers $N$ capillaries, that do not intersect and run with the same orientation. The capillaries are confined within a cylindrical model of length $L$ (m) and radius $R$ (m). They are permitted to have different radius $r$ (m), but a single length $l$. Thus, the tortuosity $\tau = l/L$ (–) is an effective property. Each tortuous and constrictive capillary presents a varying



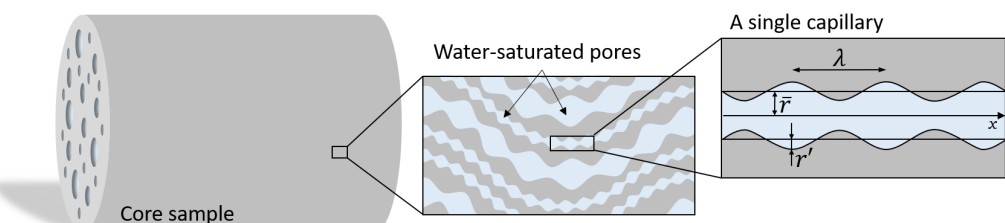

**Figure 2.** The porous rock model is composed of a large number of sinusoidal and tortuous capillaries in the cylindrical Representative Elementary Volume (REV) corresponding to the core sample. All the capillaries have the same tortuous length and their radii follow a fractal distribution. The considered pore geometry corresponds to the one from Guarracino et al. (2014) : $\bar{r}$ is the average pore radius while $r'$ is the amplitude of the sinusoidal fluctuation, and $\lambda$ is the wavelength.

radius $r(x)$ (m) defined with the following sinusoidal expression,

$$r(x) \;=\; \bar{r} + r'\sin\left(\frac{2\pi}{\lambda}x\right) \;=\; \bar{r}\left(1 + 2a\,\sin\left(\frac{2\pi}{\lambda}x\right)\right), \tag{7}$$

where $\bar{r}$ (m) is the average capillary radius, $r'$ (m) the amplitude of the radius size fluctuation, and $\lambda$ (m) is the wavelength. The

parameter $a$ (–) is the pore radius fluctuation ratio, also called the constriction factor. It is defined by $a = r'/2\bar{r}$, thus ranging from 0 to 0.5, corresponding to cylindrical pores ($r(x) \;=\; \bar{r}$) and periodically closed pores, respectively.

From Jackson's formulation based on capillary radius (Jackson, 2008), the number of capillaries is expressed as

$$N = \int_{\bar{r}_{min}}^{\bar{r}_{max}} n(\bar{r})d\bar{r}, \tag{8}$$

with $n(\bar{r})d\bar{r}$ the number of capillaries of radius between $\bar{r}$ and $\bar{r} + d\bar{r}$. As many geologic materials present a skewed pore

size distribution (e.g., Bennion and Griffiths, 1966; Dullien, 1992), we use a pseudo-fractal law to describe the distribution of capillaries throughout the model (Jackson, 2008; Vinogradov et al., 2021)

$$-dN = n(\bar{r})d\bar{r} = D\left(\frac{\bar{r} - \bar{r}_{max}}{\bar{r}_{min} - \bar{r}_{max}}\right)^{m_j}, \tag{9}$$

where $0 < m_j < \infty$ is a dimensionless exponent. $\bar{r}_{min}$ and $\bar{r}_{max}$ (m) are the extreme average pore sizes. They are estimated from the chord length computation using 3D tomography of the samples before and after the dissolution experiments Thus,

$N$ is calculated for average pore sizes $\bar{r}$ comprised between these two extremes. The normalization coefficient $D = 1$ since the pore size distribution from the chord lengths estimation is normalized and expressed in percent. The minus sign of $-dN$ implies that the number of pores decreases when the average radius increases (Soldi et al., 2017; Thanh et al., 2019; Yu et al., 2003).



### 2.3.2 Electrical properties

We follow the approach of Rembert et al. (2020), which consists in the expression of the electrical properties at the pore scale before upscaling them considering the bundle of capillaries as an equivalent circuit of parallel conductances. This approach is valid for a negligible surface conductivity and leads to

$$\sigma = \frac{\sigma_w \phi (1 - 4a^2)^{3/2}}{\tau^2 (1 + 2a^2)} = \frac{\sigma_w \phi f}{\tau^2}, \tag{10}$$

where $f$ (–) is the constrictivity of the porous medium varying from 0 to 1, corresponding to periodically closed capillaries and
cylindrical capillaries, respectively. This approach quantifies the effect of pore throats and pore bodies on the medium electrical conductivity.

Neglecting the surface conductivity in Eq. (2), Eq. (10) leads to the following definition of the formation factor

$$F = \frac{\tau^2}{\phi f}. \tag{11}$$

Rembert et al. (2020) demonstrated that the constriction factor and the tortuosity vary during dissolution process, since it affects
the structure of the porous medium. Thus, $a(\phi)$ and $\tau(\phi)$ are defined following logarithmic laws

$$a(\phi) = -P_a \log(\phi) \tag{12}$$

and

$$\tau(\phi) = 1 - P_\tau \log(\phi), \tag{13}$$

where $P_a$ and $P_\tau$ are empirical dimensionless parameters.

### 2.3.3 Johnson length

The Johnson length $\Lambda$ (m) represents a characteristic length that corresponds to the dynamically connected pores (Banavar and Schwartz, 1987; Ghanbarian, 2020). Following Johnson et al. (1986), it can be defined from the permeability and the formation factor by

$$\Lambda = \sqrt{8kF}. \tag{14}$$

Numerical studies have shown that calcite dissolution can strongly modify this characteristic length $\Lambda$ (Niu and Zhang, 2019).

### 3 Results and Discussion

### 3.1 Hydro-chemical and electrical results from percolation experiments

Figure 3 displays the evolution over time of measured and calculated parameters for the two percolation experiments, with red curves for E04 and blue curves for E05. In detail there are porosity, permeability, outlet pH values, calcium concentration and





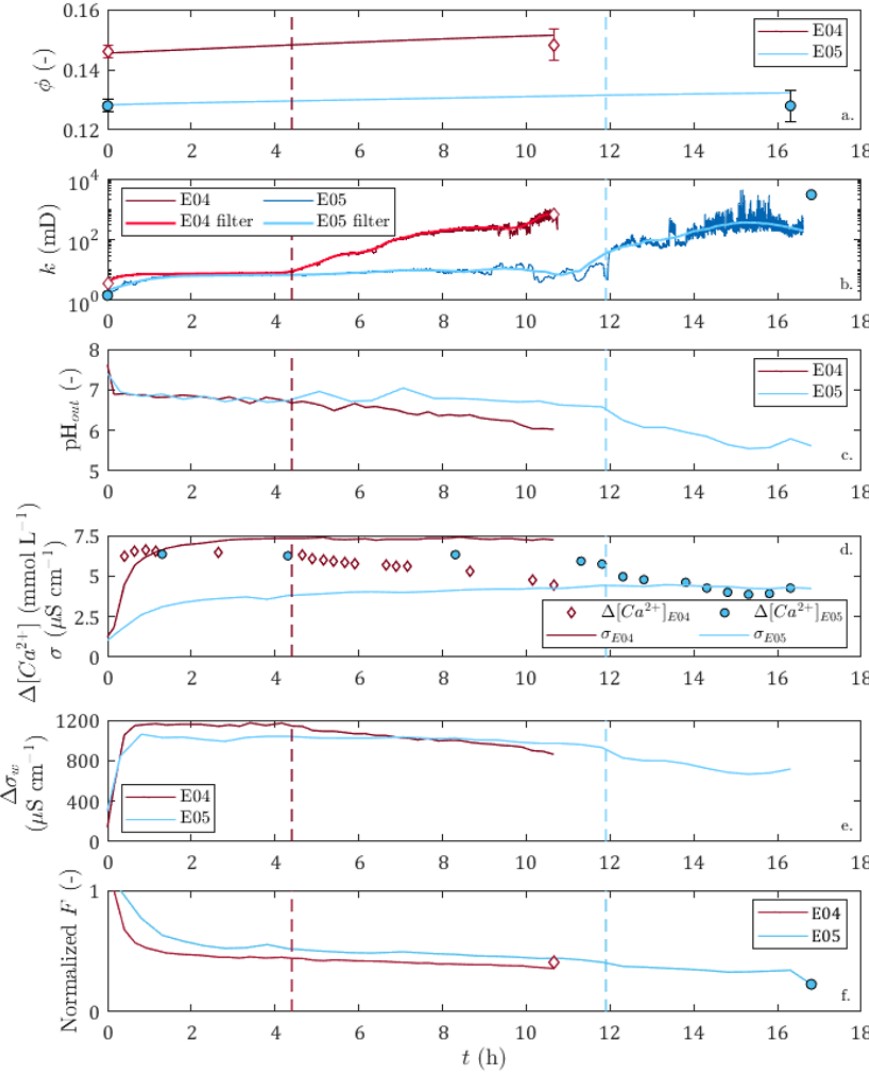

**Figure 3.** Time variations for E04 and E05 dissolution experiments of (a) the porosity, (b) the permeability, (c) the pH of the outlet sampled solution, (d) the differential calcium concentration between the inlet solution and the sampled outlet solution, and the rock sample electrical conductivity, (e) the differential electrical conductivity between the inlet solution and the sampled outlet solution, and (f) the formation factor. The red and blue vertical dashed lines correspond to the percolation times of E04 and E05, respectively. Apart from calcium data points, diamonds and circles represent the initial and final porosity with error bars, permeability, and formation factor values obtained from petrophysical measurements for E04 and E05, respectively.

core sample conductivity, water conductivity, and normalized formation factor. The initial and final values of the measured parameters obtained from petrophysical measurements are presented in Table 1. They are also represented for the initial and





final porosity with error bars, permeability, and normalized formation factor values as diamonds and circles for E04 and E05, in Fig. 3, respectively. One can note that error bars are higher for final porosity values compared to the initial values. This comes from the formation of conduits through the sample due to dissolution, which allows water to leak out and is responsible

for the measurement accuracy deterioration.

**Table 1.** Petrophysical properties from laboratory measurements of samples at initial and final states, i.e., before and after the percolation experiments with porosity $\phi_L$ (%), permeability $k_L$ (mD), formation factor $F_L$ (–), cementation exponent $m_L$ (–) and electrical tortuosity $\tau_e$ (–). The index $L$ stands for Laboratory measurements and $\phi_{ch}$ (–) is the porosity from chemistry calculations.

| | Samples | $\phi_L$ | | $k_L$ | $F_L$ | $m_L$ | $\tau_e$ |
|---|---|---|---|---|---|---|---|
| Initial | E04 | 14.6 | | 3.77 | 103.1 | 2.41 | 3.87 |
| | E05 | 12.8 | | 1.47 | 222.2 | 2.63 | 5.34 |
| | Samples | $\phi_L$ | $\phi_{ch}$ | $k_L$ | $F_L$ | $m_L$ | $\tau_e$ |
| Final | E04 | 14.8 | 15.1 | 700 | 41.5 | 1.95 | 2.48 |
| | E05 | 12.8 | 13.2 | 3000 | 50.3 | 1.90 | 2.53 |

One can observe that E04 and E05 curves present similar amplitudes and shapes for all the measured and calculated parameters displayed in Fig. 3. However, due to lower flow regime (Leger et al., 2022a), the curves of E05 present retarded or lower variations compared to E04 curves.

The generation of dissolved calcium (Fig. 3d) and the increase of porosity (Fig. 3a), permeability (Fig. 3b), and water

conductivity difference (Fig. 3e) indicate that calcite dissolution is the main chemical reaction for both experiments. However, calcium difference shows slow decrease until the end of the experiment, which can also be seen in the pH curves and the water conductivity difference curves (Fig. 3c and e), and indicates that the dissolution regime was initially strong and got weaker over time.

The permeability curves (Fig. 3b) show a small initial increase followed by a stabilization for a few hours, then they increase

sharply and reach final values several orders of magnitude higher than the initial values. The initiation of this sharp increase in permeability is interpreted as being due to the breakthrough of the sample by the acid solution, and the corresponding time, defined as the percolation time, is highlighted as a vertical dashed line in Fig. 3 (in red for E04 and blue for E05). The percolation times are 4.3 and 11.9 h for E04 and E05, respectively. For E05 the percolation time is also observable through a slight break in slope on the pH, dissolved calcium, and water conductivity curves.

Before the percolation time, one can observe that E05 permeability curve presents variations attributed to complex effects of grain displacement and transitory pore clogging. After percolation time, both E04 and E05 permeability curves present high-frequency noise corresponding to the limit of detection of the sensors of differential pressure. In order to avoid these instabilities, the curves are filtered. After a comparison with the moving average and Butterworth filters, we choose the Savitsky-Golay filter, since it offers a better adjustment. We observe that the filtered curves are well superimposed on the curves of

the measurements, except between 10 and 12 h for the dissolution of E05, because of the transient complex effects. The final permeability of E05 is higher than the last value of the temporal series. This discrepancy is also due to the limit of detection of





the sensors of differential pressure. However, we did not take this higher value into account when filtering, to be in accordance with the temporal series.

The sample conductivity increases during both experiments (Fig. 3d). At first glance, this increase could be attributed to
the increase of the pore water conductivity (Fig. 3e) instead of calcite dissolution. However, one can observe that the normalized formation factor (Fig. 3f) decreases during the experiment. Thus, the sample conductivity and formation factor variations are linked to calcite dissolution. It can be observed that for sample conductivity and formation factor, the variations are monotonous, with a rapid initial change followed by a gentler slope until the end of the experiment. The initial slopes are correlated to higher calcium concentrations at the beginning of the experiment (Fig. 3d) and are, thus, due to initial strong
dissolution regimes. The sample conductivity and formation factor smooth variations show that they are more impacted by porosity increase than by permeability variations (Fig. 3a and b). Furthermore, the comparison of the results between E04 and E05 shows that with equal water conductivity and close permeability values before the breakthrough, higher porosity induces higher sample conductivity. Compared to the petrophysical measurements, the normalized formation factor curves show good agreement with the measured amplitude difference between the initial and final values and represented as the final normalized
formation factor (Fig. 3f).

### 3.2 Applying the pseudo-fractal law to the chord length distribution

Leger et al. (2022a) have published the chord length distribution of E04 and E05 samples before and after percolation in X, Y, and Z directions. Since there are some slight variations depending on the direction, we apply the pseudo-fractal law describing the distribution of capillaries using Eq. (9) on the mean chord length distribution. The adjusted parameter $m_j$ is determined
using a Monte-Carlo optimization scheme. From the results displayed in Fig. 4, the pseudo-fractal law shows a good agreement with the chord length distributions for both samples.

In order to attest of the accuracy of the pseudo-fractal law to describe the chord length distribution, we computed the corresponding error $\epsilon < 0.5\,\%$ for each value of the exponent $m_j$. The error $\epsilon$ is defined as follows

$$\epsilon = \frac{1}{N^d} \sum_{i=1}^{N^d} |n_i^d - n_i^m|, \qquad (15)$$

where $N^d$, $n_i^d$, and $n_i^m$ refer to the number of data, the number of capillaries from the data, and the number of capillaries from the model, respectively. The best adjustment is found for values of $m_j$ of the same order of magnitude (see legends of Fig. 4), reflecting that E04 and E05 are samples of the same rock type. However, since $m_j$ can vary over large ranges, the slight differences obtained for the four different data sets (although accurately determined, see Appendix A), cannot be interpreted as impact of the dissolution on this parameter describing the pore size distribution, while percolation has a huge impact on the
rock properties as presented above (see Fig. 3 and Table 1). Thus, the evolution of $m_j$ is not a sufficient indicator of the effect of dissolution on the sample.

Moreover, Leger et al. (2022a) have also determined the pore size distribution by centrifugation. Since the size intervals for the determination of the pore number are not regular, we could not use the pseudo-fractal law on these data. Nevertheless, the





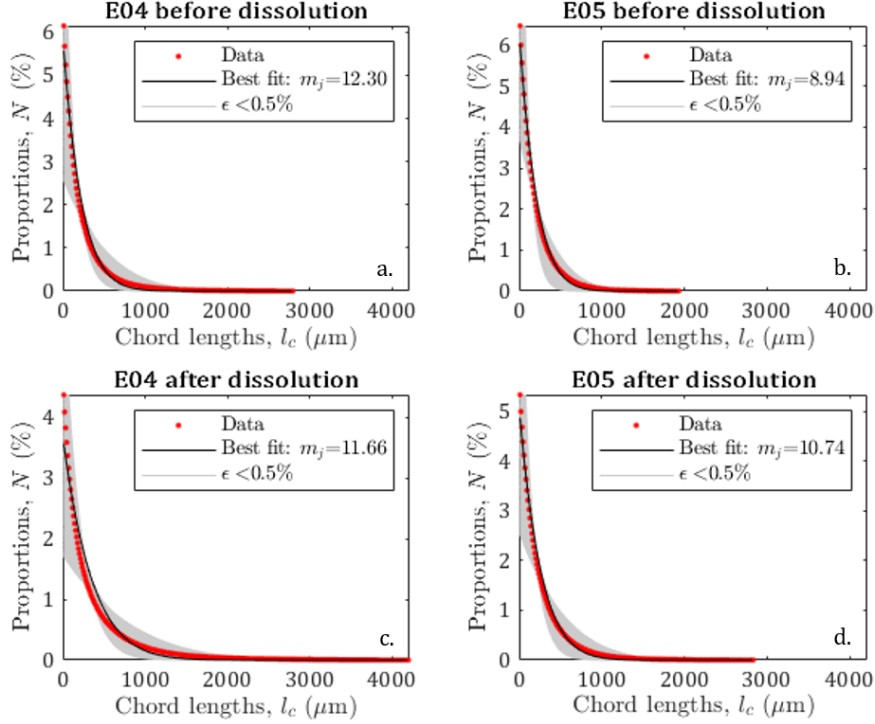

**Figure 4.** Pore size distribution of E04 and E05 core samples before and after dissolution from the chord length estimation (red curve), from the pseudo-fractal law (black curve corresponds to the best adjustment of the power law exponent and gray curves correspond to the pseudo-fractal fit with an error of less than 0.5%.

characterization of pore size distribution by centrifugation has shown that between 46 and 70 % of the pores have a pore size

comprised between 0.5 and 12 μm, while the minimal resolution of the chord length is of 12 μm. In addition, a very porous area of the sample only composed of small pores can be interpreted by the tomography software as a single pore over the entire pixel. This means that the chord length distribution has not the best precision for the smallest pores distribution. However, Leger and Luquot (2021) noticed the pore size distribution from the two methods are similar and consistent, leading to the conclusion that large pores are more likely to control rock parameters on core scale. In addition, this importance of macropores

is even more of interest in reactive transport, as it can be seen in fractured media (e.g., Noiriel, 2015; Garcia-Rios et al., 2017).

### 3.3 Influence of rock structure on electrical signature

We have shown in the previous section that the pore size distributions of both samples were successfully describable using pseudo-fractal laws. Thus, we can apply the relationship relating the formation factor and the porosity developed in Eq. (11). We can see in Fig. 5 the results of the Monte-Carlo inversions to adjust the constrictivity $f$, defined from Eq. (10), and the





tortuosity $\tau$. To reflect the structural changes in the porous medium, we consider the parameters as distributions $a(\phi)$ and $\tau(\phi)$, expressed in Eq. (12) and Eq. (13). We observe that the evolution of $f$ and $\tau$ with the porosity is physically plausible as the increase of porosity due to calcite dissolution reflects less complex medium geometries (i.e., less constrictive and less tortuous), described with higher values of $f$ and lower values of $\tau$. The accuracy of the model is evaluated using the error computation of Eq. (15). For both samples it is below 10 %.

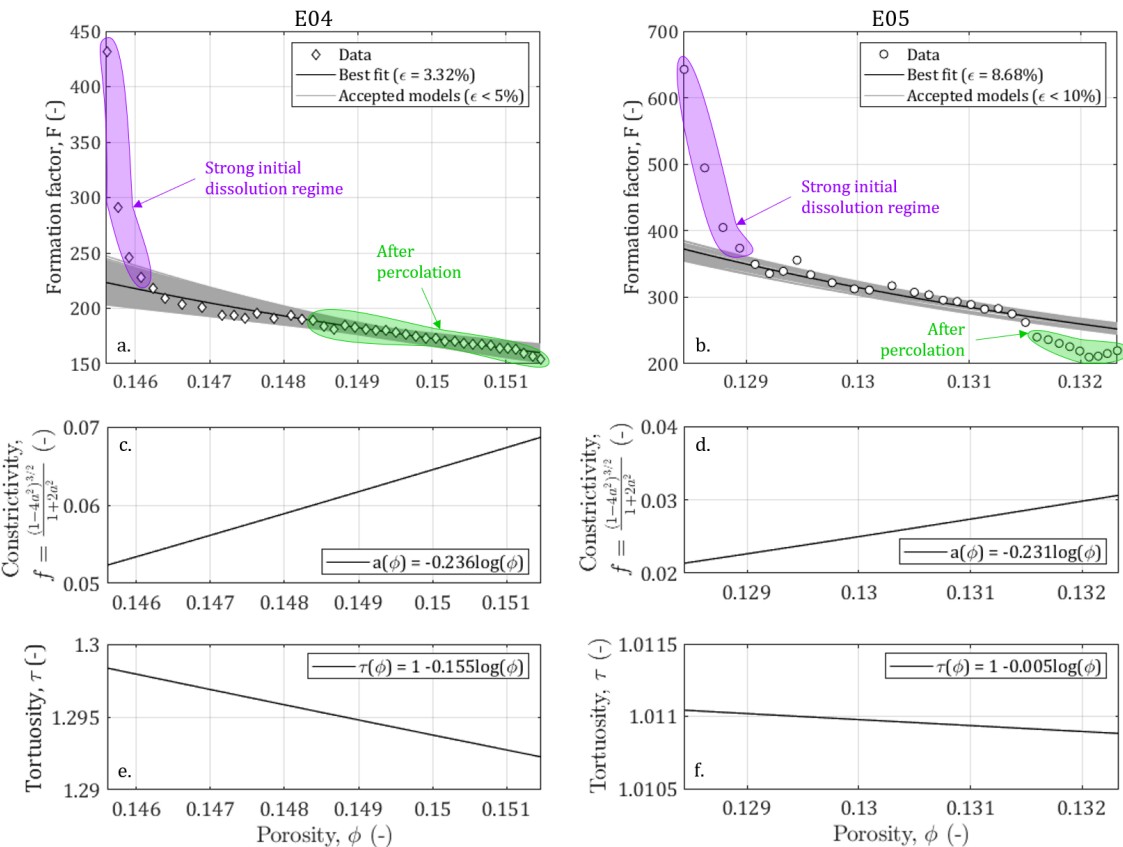

**Figure 5.** Results of the Monte-Carlo modeling of the relationship between the formation factor and the porosity for samples E04 and E05 affected by calcite dissolution and evolution of the constrictivity and the tortuosity in function of the porosity increase.

The constrictivity varies over different ranges of values between E04 and E05 (Fig. 5c and d). For E05, $f$ is lower by half (i.e., more constrictive) compared to E04, which is in agreement with E04 higher porosity values. In contrast, the evolution with porosity follows roughly the same law for these two samples. This indicates that we are working on two samples of the same rock type with similar pore shapes.

The tortuosity varies little during the experiment for both samples, although it is more pronounced for E05 (Fig. 5e and f).
A surprising result is that the tortuosity values are higher for E04 than for E05, while E05 is less porous. This result is in





contradiction with the values of initial and final electrical tortuosity $\tau_e$ displayed in Table 1, showing higher values for both samples and higher values for E05 compared to E04. This discrepancy has already been observed in Niu and Zhang (2019) and can be explained with the definition of $\tau_e$, given in Eq. (3), which combines the impact of the tortuosity and the constrictivity and is, thus, artificially higher. Moreover, the main conduits generated by dissolution can be visualized from tomographic

images. Figure 6 shows the phase connecting the inlet to the outlet for the two samples, according to two different image treatments. Two segmentations based on a region growing method were applied to isolate two different void phases. The red phase represents the phase connecting the largest pores in the samples from the inlet to the outlet, as already explained and used in Leger et al. (2022a). The blue phase only shows the darker pixels of the images, corresponding to the center of the conduit used by the fluid to cross the samples. In comparison E05 has a straighter path than E04. Thus, the model parameters

adjustment reproduces well the tomography observations and the experimental data.

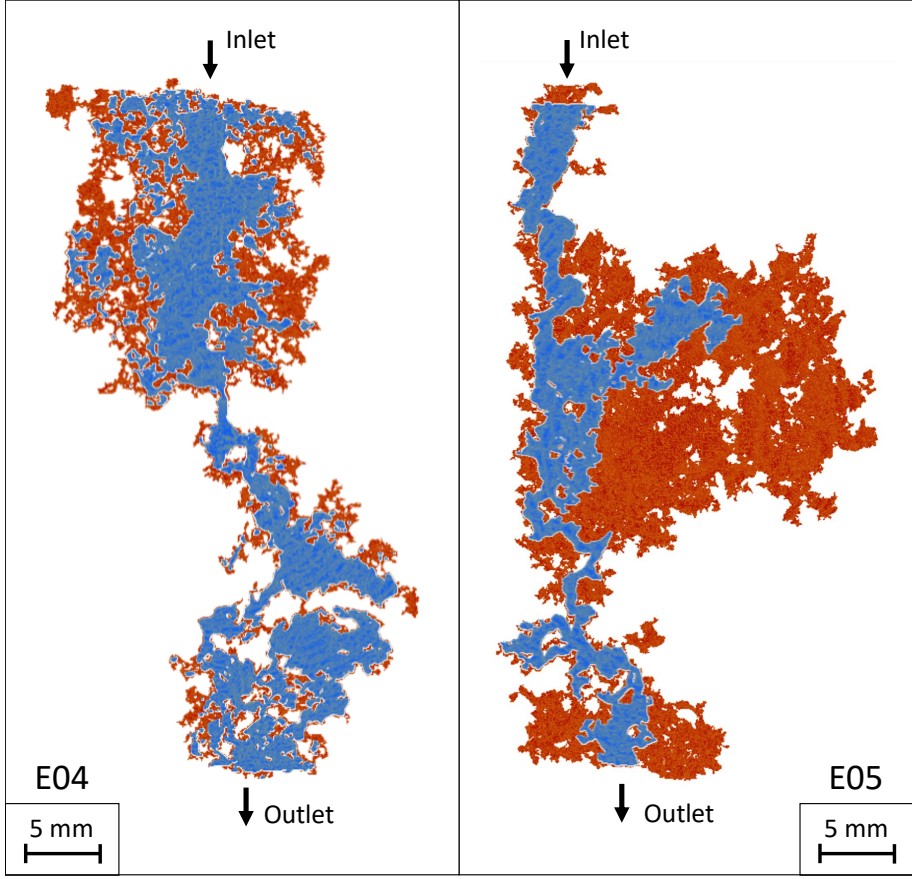

**Figure 6.** Main path conduits of E04 and E05 opened due to calcite dissolution. The samples are displayed with the acid injection from the top to the bottom. The red phase shows the complete connected phase and the blue phase shows a restricted phase where the flow concentrates.





However, for the lowest porosity data in Fig. 5, the formation factor shows high values, highlighted in purple, that do not fit with the gentle slope of the model. As mentioned earlier, these points correspond to an initial strong dissolution regime that is out of the validity domain of the model (Guarracino et al., 2014; Rembert et al., 2020). Moreover, the points in green in Fig. 5 highlight the data acquired after the percolation, where the samples cannot be considered anymore as representative
elementary volume (REV) and where the model is not valid neither. Nevertheless, for E04, the acid percolation does not show radical changes in the formation factor and porosity compare to the smooth tendency that follows the initial strong dissolution regime, whereas for E05, the formation factor falls out of range after percolation. These different behaviors are due to the imposed hydrodynamic regimes and their consequences on the dissolution profile. Indeed, the red phase in Fig. 6 shows that the new paths created in E04 are concentrated in a reduced volume of the sample, while for E05 the red phase is extensively
spread in the upper part of the sample. This portion of the sample, close to the inlet, was probably dissolved at an early stage of the experiment, therefore, creating no apparent effect on the permeability or formation factor before the sample percolation.

### 3.4 Influence of hydrodynamic properties on rock structure and electrical signature

#### 3.4.1 Impact of the experimental conditions and permeability changes

In this section, we are interested in highlighting the link between the evolution of hydrodynamic and electrical properties. The permeability is plotted as a function of the formation factor and porosity in Fig. 7. We can first note that the permeability varies over the same ranges of values for both samples, while E04 has a larger porosity (and lower formation factor) compared to E05. As discussed in the literature (e.g., Guarracino et al., 2014; Soldi et al., 2017; Rembert et al., 2020), the pore throats play a primary role on the hydrodynamic behavior of porous media. According to the model used in this study, the studied samples
follow the same distribution for constrictivity (Fig. 5), which is consistent with the similarity in permeability.

It can be observed that before the breakthrough, the permeability increases little, while the formation factor and the porosity vary strongly. Then, after percolation, the trends are reversed, the permeability increases strongly compared to the lower variation of the porosity and the formation factor. This behavior change is very clear for both samples and indicates that monitoring electrical properties allows us to be sensitive to the impact of dissolution on the porous medium long before the
sample is percolated.

In order to locate our experiments in the scientific landscape, we plotted our data with the well-known power-law relating permeability and porosity $k \sim \phi^n$ (Kozeny, 1927; Carman, 1937), where $n$ (–) is an empirical exponent. This relationship has been widely used in literature, sometimes by trying to determine the $n$ exponent (Martys et al., 1994), sometimes by taking into account the critical pore size (Gueguen and Dienes, 1989; Thompson et al., 1987), the specific surface with the Kozeny
factor (Paterson, 1983; Walsh and Brace, 1984; Fabricius et al., 2007; Weger et al., 2009), or the grain diameter in a non-fractal (Kozeny, 1927; Carman, 1937, 1948, 1956; Bourbie et al., 1987) or in a fractal dimension (Pape et al., 1999; Costa, 2006). The classic power-law has been tested here and the $n$ values obtained are a bit high, but are still consistent with those proposed in





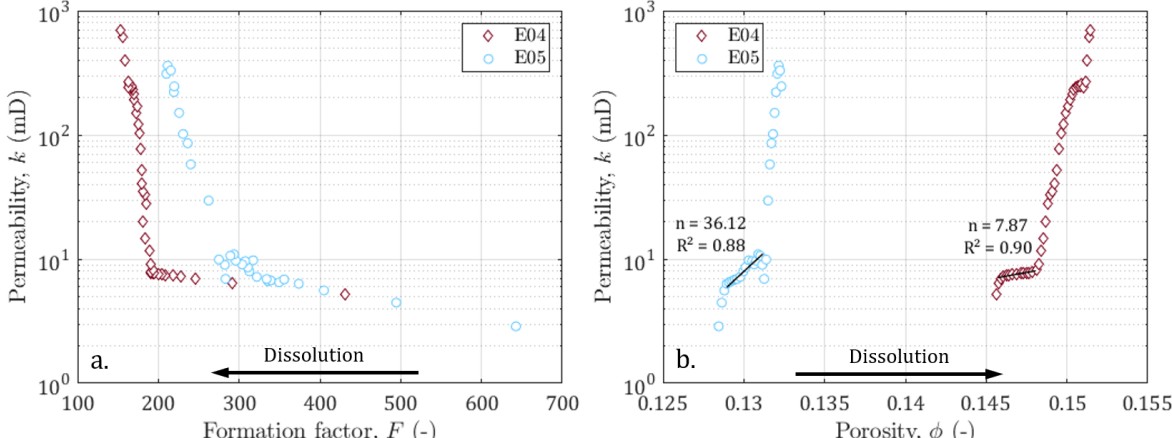

**Figure 7.** Permeability against (a) formation factor and (b) porosity. Black arrows indicate the direction of the impact of dissolution on the formation factor and porosity. Black lines and annotations indicate the slopes, the exponent, and the coefficient of determination of the Kozeny-Carman equation for the data before percolation.

the literature on carbonate rocks, known for their natural heterogeneity and also the one created by dissolution (see Leger et al., 2022b, for a synthetic review).

### 3.4.2 The characteristic pore size defined through the Johnson length

The comparison of the permeability of E04 and E05 reveals that they are samples with similar initial hydrodynamic properties related to their identical rock type (Table 1). However, their difference of porosity leads to different ranges for the formation factor, and the different imposed hydrodynamic regimes lead to higher final permeability for E05. Moreover, the use of a normalized formation factor (Fig. 3) enables to compare trends but does not allow to quantitatively interpret the electrical monitoring in term of rock structure. Thus, as recommended by Niu and Zhang (2019), we use the Johnson length as an effective pore size (Fig. 8), that enables to compare the behavior of both samples E04 and E05 regarding other data sets from the literature (Niu and Zhang, 2019; Vialle et al., 2014).

The series from Niu and Zhang (2019) comes from calcite dissolution simulation of a digital image of a carbonate mudstone (Wellington Formation, Kansas, USA), that has an initial porosity of 0.13. The fluid transport is assumed as advection dominated ($Pe > 1$), and under this transport condition, referring to Pereira Nunes et al. (2016), they impose a transport-limited case, related to wormholing, in which the reaction at the solid-liquid interface is limited by the diffusion of reactants to and from solid surfaces ($PeDa > 1$, where $Da = k_r/u$ is the so-called Damköhler number, and $k_r$ is the local reaction rate).

The data points from Vialle et al. (2014) correspond to bioclastic limestone (Estaillades limestone, France), composed of more than 99% calcite, and of initial porosity of 0.29. The values for $Pe$ ranges from 18.5 (for the smaller grains) to 37 (for the



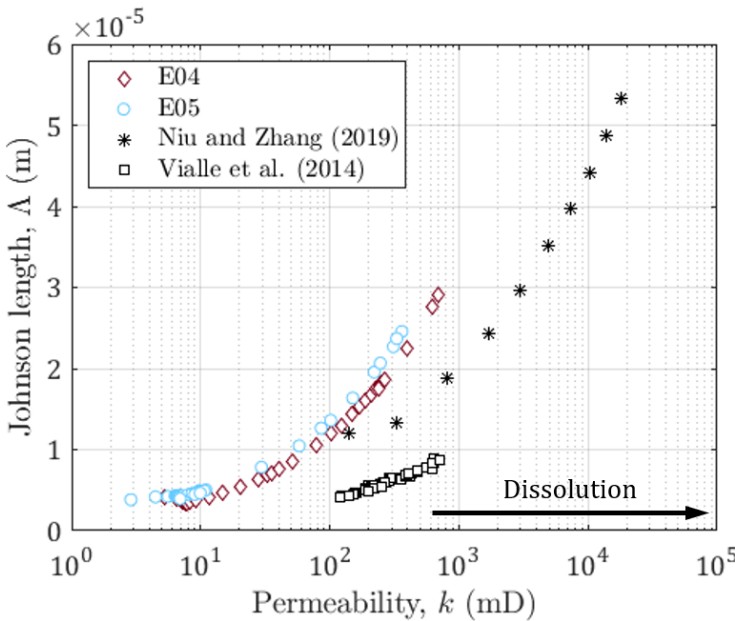

**Figure 8.** Johnson length against the permeability for E04, E05, and data sets from Niu and Zhang (2019) and Vialle et al. (2014). The black arrow indicates the direction of the impact of dissolution on the permeability.

larger ones) and $Da = 3486$. Meaning that for this data set, the dissolution is also transport controlled and leads to dominant wormhole formation.

One can observe that all the series present the same behavior, but with different trends and on different ranges. These differences do not seem related to initial porosity value or hydrodynamic regime. Indeed, for different Péclet numbers, there are close values and trends for E04 and E05 compared to the other series. In addition, while the carbonate mudstone from Niu and Zhang (2019) presents similar initial porosity value with E05, they do present contrasted Johnson lengths. Thus, the Johnson length is related to the rock type and highlights the difference in the pore structure among the samples.

## 4    Conclusions

This study investigates the impact of calcite dissolution on conductivity monitoring through percolation experiments of two crinoidal limestone core samples in different hydrodynamic conditions. We, thus, demonstrate that lower the Peclet condition applied to the samples is, lower the dissolution rate is.

Then, we express the electrical conductivity and the formation factor of the porous medium in terms of effective petrophysical parameters: tortuosity and constrictivity. In this representation, the pore space is conceptualized as a fractal cumulative distribution of tortuous capillaries with a sinusoidal variation of their radius (i.e., periodical pore throats). We verify the assumption of a fractal pore size distribution using the chord length distribution computed from XRMT images. The confrontation



of the model with the formation factor monitoring for both samples shows that the model can reproduce structural changes linked to calcite dissolution by setting logarithmic dependence of constriction factor and tortuosity to porosity. These empirical relations allow the model to accurately fit the data within the range of validity of the model, defined as a steady reaction rate obtained after first acid invasion through the sample and before the acid breakthrough. We observe that dissolution leads to less constrictive rather than less tortuous sample, thus, designating constrictivity as a property of the pore structure highly impacted

by chemical processes.

By means of an upscaling procedure, we also link the formation factor to permeability through the Johnson length. We show on the experimental data of this study and other data sets (experimental and numerical) from the literature that the Johnson length is a relevant parameter that allows comparison of chemical reaction in comparable Peclet-Damköhler conditions on different rock types (crinoidal limestone, carbonate mudstone, and bioclastic limestone).

We believe that the present study contributes to a better understanding of the links between the electrical conductivity measurement and the evolution of the hydrodynamic and structural properties of the porous medium subjected to reactive transport.

*Data availability.* The data used in this study are available in the Zenodo repository (https://zenodo.org/record/6908522).

*Sample availability.* Samples of the same rock type are available upon request.

**Appendix A: Accuracy of the determination of the pseudo-fractal exponent**

The accuracy of the pseudo-fractal computation for the determination of the pore size distribution is controlled through the error $\epsilon$ defined in Eq. (15). Since dissolution does not generate notable variations of the pseudo-fractal exponent $m_j$, we confront the computed error to a range of value $m_j$ in Fig. A1. We obtain V-shaped curves for each data set, showing that $m_j$ is well-determined for each sample, before and after percolation experiments.

*Author contributions.* Conceptualization, F.R., M.L., D.J., and L.L.; Data curation, F.R. and M.L.; Formal Analysis, F.R., Funding acquisition, F.R. and L.L.; Investigation, M.L.; Methodology, F.R., M.L., D.J., and L.L.; Project administration, L.L.; Resources, D.J. and L.L.; Software, F.R., M.L., and D.J.; Supervision, D.J. and L.L.; Validation, M.L.; Visualization, F.R. and M.L.; Writing – original draft, F.R.; Writing – review & editing, F.R., M.L., D.J., and L.L.

*Competing interests.* No competing interests are present.





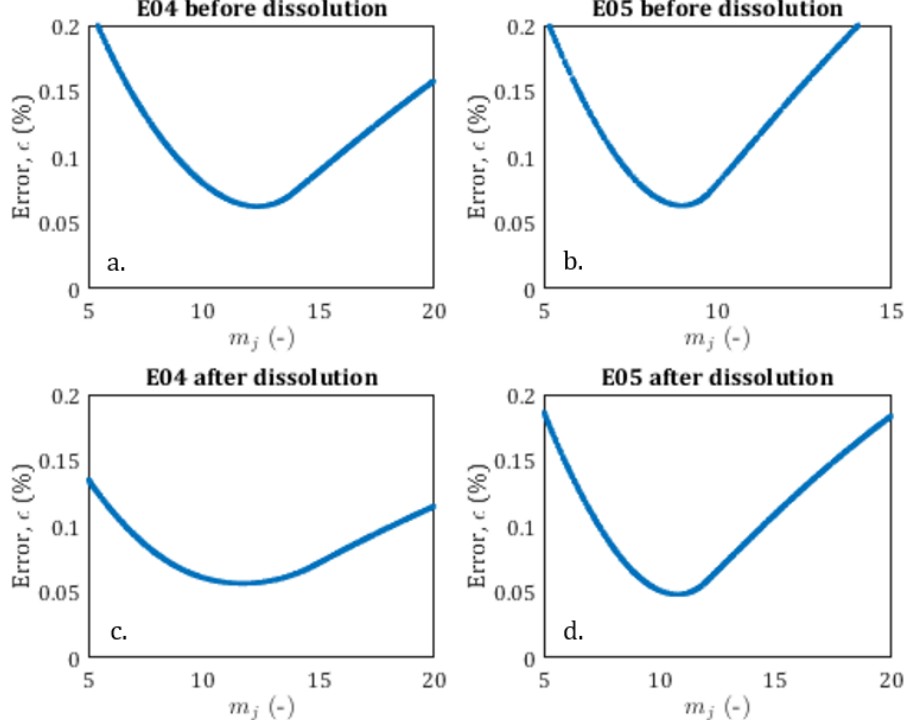

**Figure A1.** The error of the pseudo-fractal law against the exponent $m_j$.

*Acknowledgements.* This research was funded by the research projects EC2CO (CNRS-INSU) StarTrek and JPI-Water UrbanWat from WaterWorks2017, by the LabEx VOLTAIRE (ANR-10-LABX-100-01), and by a CIFRE PhD fellowship provided by Voxaya and ANRT. We acknowledge the MRI platform member of the national infrastructure France-BioImaging supported by the French National Research Agency (ANR-10-INBS-04, "Investments for the future"), the labex CEMEB (ANR-10-LABX-0004) and NUMEV (ANR-10-LABX-0020). We also acknowledge the petrophysics plateform members of Geosciences Montpellier Laboratory.



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
