# Peer review of "Geoelectrical and hydro-chemical monitoring of karst formation at the laboratory scale"

_EGUsphere, 2022_

## Author Comment (AC1)

**Response to Anonymous Referee #1**

**General comments**

In the present paper, the authors investigate the impact of calcite dissolution on the evolution of rock structure, hydrodynamics and electrical properties, as well as their coupling. Their approach consists in monitoring the permeability, calcium concentration in outlet fluid (to infer changes in porosity) and electrical conductivity of the sample and outlet fluid during the injection of acid brine into two limestone samples with different flow rates (and Peclet numbers). The samples are also characterized using traditional petrophysical lab measurements as well as micro-CT imaging before and after experiments. The electrical conductivity data are interpreted by using a physics-based model in which the pore space is represented by a fractal cumulative distribution of sinusoidal tortuous capillaries, and then compared with data available in the literature (Niu and Zhang, 2019; Vialle et al., 2014) by using the Johnson length. The results are well described and supported by clear and effective figures. However, they should be discussed in more depth, especially the ones that are unique to this paper. The laboratory and micro- CT petrophysical characterization of the samples before experiments are reported in a previous publication (Leger et al., 2021). The evolution of porosity and permeability during acid injection into the samples and the comparative characterization of petrophysical properties using lab and micro-CT techniques are also the scope of a previous paper (Leger et al.. 2022a). The monitoring of electrical conductivity during these experiments and their interpretation is the novel aspect presented in this third paper. The objective of the paper as stated by the author is to give some insights on how the electrical signal is impacted by calcite dissolution, yet the conclusion does not clearly open back up to that broader goal. In general, the title of the paper is misleading, a few important details are missing in the material and methods section, some sections lack clarity and a few critical elements should be discussed or at least mentioned in the paper before it can be considered for publication.

The authors would like to thank the reviewer for his detailed reading of this study and related articles, as well as for his careful highlighting of the strengths of this article. The authors carefully acknowledge the points for further study and clarification raised by the reviewer and detailed in his or her comments below. The authors hope that the point-by-point responses will be satisfactory.

**Specific comments**

1. The title of the paper is "Geoelectrical and hydro-chemical monitoring of karst formation at the laboratory scale". The abstract indicates the need for "a better understanding of the mechanisms responsible for conduits formation in the rock mass and the development of detection methods for these hydrological and geochemical processes". In the short summary the authors states that their work is "showing the benefits of geoelectrical monitoring to map karst formation". I found this to be quite misleading. One would expect the results to be discussed in terms of geophysical monitoring of karst formations at the field scale: would similar hydro-geochemical trends be expected for a much less acidic fluid? How much dissolution should occur before it can be identified on ERT data (presenting electrode spacing that would be much larger than in the experiments presented in the paper, hence lower resolution)? What would be the order of magnitude of corresponding time-scales? In general I would suggest removing "karst formation" from the title and change it with "calcite dissolution". The introduction does not even have a paragraph dedicated to karsts. It goes from carbonate formations to dissolution of carbonate rocks. Moreover the authors do not reproduce a karst at the laboratory scale. Instead they study limestone rock samples (which would correspond to the matrix component of a karst, in its saturated zone, far from fractures and conduits). The injected fluid is also very different from

water that would flow through a karst. The application is almost more relevant to acid injection to increase formation permeability in the vicinity of a borehole. I would also refrain from using "conduits" in this work, as for the karst hydrology community this mostly refers to the larger dissolution features, for which Darcy's law is not applicable as the flow tends to be turbulent (open channel/pipe flow). I would suggest using "porosity development in the rock matrix" or simply "dissolution features", but avoid conduits.

The point raised by the reviewer about the reader's expectations based on the title of this article is interesting and is of particular concern to the authors. As mentioned by the reviewer, the results of this study are not directly transferable to the study of karsts in the field but do help to weave together connections between scales about the relevance of geoelectrical monitoring to the monitoring of karst systems. Indeed, the main result of this study is the evidence of a significant temporal variation in the electrical measurement well before the acid percolation, translated by a jump of the permeability, blind to this formation by the dissolution of a main porous channel before the total percolation. Thus, it appears essential for the authors to emphasize this link with the dynamics of karst networks. The authors propose for the revised version of the article to adjust the title of the article, as suggested by the reviewer, but to keep the link with the survey of karsts in the abstract. Furthermore, the authors take note of the vocabulary suggestion made by the reviewer and will modify the manuscript accordingly by adding to the examples given by the reviewer, expressions such as "wormhole formation".

2. This paper presents results obtained during dissolution experiments conducted on two samples, that were already the scope of two previous papers: one centering on multimodal petrophysical and structural characterization of the samples (Leger et al., 2021), and one focusing on the evolution of porosity and permeability of the samples during the dissolution experiments (Leger et al., 2022a). The key findings of these papers should be dedicated a paragraph in the introduction before introducing the goal of the present paper.

The authors strongly agree with the reviewer about adding in the introduction a paragraph which emphasizes the key findings from the related articles from Leger et al. (2021) and Leger et al. (2022a).

3. The materials and methods section needs have the following missing information and descriptions that need to be more detailed and/or better justified

The authors are very thankful to the reviewer for making such meticulous remarks about the materials and methods section, which will gain clarity.

- Instrument used for XRF (lines 60). If you specify for XRD then also specify for XRF, otherwise just simplify for both and refer to the characterizarion paper "XRD and XRF measurements show that… (leger et al., 2021)"

The authors agree with the suggestion of the reviewer and, in the revised version, the authors will add that XRF is conducted on an FEI Quanta 200 FEG.

- Method used for porosity (helium porosity, water-saturated porosity?), permeability (it is gas permeability, water permeability?) (Line 62). Even if the full descriptions are available in previous papers, essential details should be reminded here.

As recommended by the reviewer and referring to leger et al. (2021), the authors will specify in the revised version of the manuscript that the petrophysical characterization was carried out on dry and saturated samples using four different fluids in chemical equilibrium with the samples at four NaCl concentrations: 0.3 mol L$^{-1}$, 0.2 mol L$^{-1}$, 0.1 mol L$^{-1}$, and 0.05 mol L$^{-1}$. Between each saturation with the different NaCl concentrations, the samples were dried in an oven. Gas porosity and permeability were measured by helium injection using a porosimeter and permeameter, respectively. Liquid porosity was measured on saturated samples using a double-weighing on saturated and dried samples, knowing the samples' dimensions. Liquid permeability was measured on saturated samples in an experimental device, knowing the differential pressure and using Darcy's law.

- The identification of red and blue phases of the pore space presented Figure 6 and discussed in section 3.3 should be introduced in the paragraph related to XRMT analysis (Lines 82-85). In particular the novel processing using two segmentations, which differs from the one presented in Leger et al. (2022a) should be detailed here. Side note: use actual numbers and not "the largest pores" or "the darker pixels", for instance "pores presenting a mean diameter comprised between…. and …" Any quantification of the percentage of the samples' pore space that are below the voxel size (12 µm) should also be indicated in this paragraph (rather than Page 12, line 11)

The red segmentation corresponds to the one presented in Leger et al. (2022a). The segmentation procedure does not allow to discretize the pore space in terms of quantified pore sizes since it is based on grayscale visualization and segregation. This is a qualitative procedure.

- In sub-section 2.2, the accuracy and detection limit of the essential components of the experimental set-up (pressure sensors, flow rate, pH, conductivity, ion concentrations…) be listed, as well as standard error on measured properties.

The remark of the reviewer concerning the accuracy and the dispersion of the data is of legitimate interest. To meet these concerns, the authors will give in the new version of the manuscript precision about the sensors used for the measurements of the different properties. pH and inlet and outlet pore water electrical conductivity were measured with JUMO sensors. Permeability is estimated from pressure difference measurements which are performed using an absolute pressure (Gems) and a differential pressure (Keller) sensor. All of these sensors were connected to a Labjack system for continuous acquisition. The constant flow rate was ensured with a peristaltic pump (Gilson, MiniPls3). All of these devices and sensors were used under normal operating conditions, leading to detection limits inherent to the equipment and available in their respective datasheets. The standard error can be determined for the rock sample's electrical conductivity and permeability on steady portions since the acquisition rate is high compared to the time variations. From this, the standard errors are estimated around 10$^{-3}$ µS cm$^{-1}$ and 10$^{-2}$ mD.

- There is no paragraph in sub-section 2.2 explaining how F is experimentally calculated. It does not have to be lengthy at all but it should be clearly written somewhere that for each time step, the formation factor (which I think is later referred to normalized formation factor or normalized F?) is calculated using equation (65), neglecting surface conductivity and taking xxx for the brine conductivity. The authors actually never explicitly mention what is taken for xxx. Is it the differential electrical conductivity between the outlet and inlet brine? Again, make sure to introduce what normalized F is before it suddenly comes up in the result section (Page 9, Line 175).

*The formation factor is calculated using the outlet values of the pore water conductivity. Since this approximation gives an apparent formation factor, not directly comparable with the initial and final values, the authors normalize the formation factor to propose a fair comparison.*

4. Figure 3. The sample conductivity is presented on the same graph as the differential calcium concentration. This is somewhat confusing. Wouldn't it be better to present it together with the brine conductivity (with two y-axis)? Or actually present the brine conductivity together with calcium concentration since an increase of ion in the brine would result in an increase in electrical conductivity? If the only change in the water chemistry is due to the calcite dissolution, you should have a straight-forward relationship in between the two (evolution of calcium concentration and evolution of brine electrical conductivity), as electrical conductivity is directly related to ionic strength. In the paragraph describing these particular graphs (Figure 3d, e and f), top of page 11, start by reminding the reader what one would expect to see: calcite dissolution should increase the formation electrical conductivity as it increases 1) pore volume and 2) pore water electrical conductivity.

*The authors have presented the results in Figure 3 in combined plots to propose a compact view. However, as raised by the reviewer, the combination of different data showing contrasting variations can be confusing for the reader. For the revised version, the authors will follow the reviewer's recommendation about putting the calcium concentration with the water conductivity difference. The presentation of the results will be modified accordingly in the text.*

5. There are some conflicting information regarding the importance of microstructure on rock properties. Page 12, line 238 "leading to the conclusion that large pores are more likely to control rock parameters on core scale. In addition, this importance of macropores is even more of interest in reactive transport, as it can be seen in fractured media (ref)". I would not use ref of studies of reactive transport in fractured media, and would be interested to actually see work backing up the control of large pores on reactive transport in porous media. In general, the smallest pores control fluid flow and even the slightest modifications of the constrictions of hydraulic flow paths (throats/smaller pores) can have a major impact on permeability. The importance of microstructure on rock properties is actually the main conclusion of Leger et al. (2021), and later in the paper (Page 150, Line 283-284) the authors emphasize this statement with "As discussed in the literature (refs) the pore throats play a primary role on the hydrodynamic behavior of porous media". This should really be discussed in further details since a portion of this work utilizes micro-CT images acquired at a fairly low resolution, hence only probing the largest pores. It should at least be addressed when presenting the red/blue phases of the connected pore space after acid injection.

*The authors thank the reviewer for highlighting this inaccuracy in the discussion to allow for the readjustment of the interpretation of the fit of the fractal pore size distribution. Indeed, the point raised by the reviewer perfectly underlines the interest in using a fractal model for this study. First, the good fit with the distribution of string lengths gives a fractal law fit on the largest pores. Then, the tendency of the fractal law to impose a majority proportion of small pores also coincides with the estimate of the pore size distribution obtained by centrifugation. As recommended by the reviewer, the reference to fractured media will be removed in the revised version of the manuscript and replaced by the above statement. Regarding the second part of the reviewer's comment, the authors will clarify that the use of low-resolution micro CT images is consistent to show the localization of the wormhole in the sample and to what extent the electrical measurement allows highlighting its more or less rectilinear appearance thanks to the fractal modeling which quantifies the effect of the pore throats.*

6. In the sub-section 3.4.2, the authors compare their data with data from two previous studies (Vialle et al., 2014; Niu and Zhang, 2019). The Peclet and Damkohler numbers are reminded for the two datasets. The Damkohler number of the present study is never given. It would be great to have an estimation of it, to be able to fully compare the three datasets. Even though they have different values of Pe and Da, would the three datasets still be located in the same dissolution regime – dominant wormhole? (Golfier et al., 2022- Figure 19)?

The authors strongly agree with the reviewer about adding an estimation of the Damköhler numbers. With the formula from Menke et al. (2015) the authors calculate $Da_{E04}=4.1 \times 10^{-5}$ and $Da_{E05}=8.9 \times 10^{-5}$. These low Damköhler values should lead to uniform dissolution, as established by Golfier et al. (2002) in their diagram. However, this result does not stand with the tomographic images. As discussed in Menke et al. (2016) and Leger et al. (2022b), the threshold values proposed in the diagram published in Golfier et al. (2002) are not applicable for heterogeneous porous media. Indeed, the rock initial heterogeneities allow for preferential paths which channelize the reacting solution and lead to a dominant wormhole dissolution regime.

Ref 1: Menke, H.P., Bijeljic, B., Andrew, M.G., Blunt, M.J., 2015. Dynamic three-dimensional pore-scale imaging of reaction in a carbonate at reservoir conditions. Environ. Sci. Technol. 49, 4407–4414. https://doi.org/10.1021/es505789f

Ref 2: Menke, H., Andrew, M., Blunt, M., Bijeljic, B., 2016. Reservoir condition imaging of reactive transport in heterogeneous carbonates using fast synchrotron tomography — effect of initial pore structure and flow conditions. Chem. Geol. 428, 15–26. https://doi.org/10.1016/j.chemgeo.2016.02.030

Ref 3: Golfier, F., Zarcone, C., Bazin, B., Lenormand, R., Lasseux, D., Quintard, M., 2002. On the ability of a Darcy-scale model to capture wormhole formation during the dissolution of a porous medium. J. Fluid Mech. 457. https://doi.org/10.1017/S0022112002007735

7. The conclusion section should make a better job to expand the findings of the study to geoelectrical monitoring in general. It is is a mere summary of the results, listed linearly rather than integrated and put back in context. What are the implications of the work? Limitations? Future work? Application to karst? Implications for field scale? On the one hand, the first paragraph of the conclusions is literally the main conclusion of Leger et al., (2022a). On the other hand, the statement in the result and discussion section (Page 15, Line 288) "This behavior (…) indicates that monitoring electrical properties allows us to be sensitive to the impact of dissolution on the porous medium long before the sample is percolated", which is an important statement, is not at all reflected in the conclusion.

The authors highly value this comment from the reviewer about emphasizing the important statements of the discussion in the conclusion to render it more effective in the revised version.

**Technical corrections**

- Page 2, Line 20: "…issues **related to** contamination, erosion…"
- Page 3, Line 26: "…well studied **in the laboratory** to understand…"
- Page 3, Line 27: "…and **their intricate coupling with** transport properties **such as** porosity and permeability…"
- Page 3, Line 28: "these experiments generally **rely on** imag**e** analysis**,** which is an accurate…"
- Page 3, Line 39-40: "highlighting their interest for non-intrusive characterization" – the interest of what/whom? The studies? The authors of the studies? The processes? – Please rephrase

- Page 3, Line 48-49: "Evolution of all these properties coupled with the initial conditions" unclear – precise what you mean exactly.
- Page 3, Line 49: "…is analyzed in order to **assess** in which proportion…" (**quantify** would work as well)
- Page 4, Line 63: "…on **samples saturated with brine with varying salinity** to determine…"
- Page 4, Line 89: "…of the medium through **Archie's law** (1942)"
- Page 4, Line 83: "…analyzed with a **in-house** software…" same for Page 5, Line 94
- Page 5, Line 87-88: Reverse the sentence: "**The same laboratory and imaging characterization was repeated on the samples after dissolution in order to estimate the impact of the core-flooding experiments on the rock properties.**"
- Page 6, Line 127: "…Vinogradov et al., **2021). It** uses…" (point missing)
- Page 6, Line 129: "…model consider N **capillaries that** do not…" (no coma) Page 7, Line 144: "…after the dissolution **experiments. Thus**…" (point missing) Page 8, Line 167-168: "it can be defined…formation factor **as**"
- Page 10, Table 1 – Title: remind Q/Pe for E04 and E05
- Page 14, Figure 6 - Title : indicate image resolution
- Page 15, Line 280: "In this **section we focus on** the link…"
- Page 15, Line 291: "**In order to compare our results with previously published data**…"
- Page 15, Line 297: "values obtained are a bit high" – avoid vague terms when describing scientific data.
- Page 16, Line 298: "known for their natural heterogeneity and also the one created by dissolution" What does "the one" refer to? Unclear, please rephrase.
- Page 17, Line 324: "We first **demonstrate that the lower the** Peclet condition…, **the lower** the dissolution rate is."
- Page 18, Line 334: "…as a property of the pore structure **that is impacted by chemical processes to a greater extent**" (I would also replace chemical processes by dissolution processes, as the data cannot be extended to other chemical processes such as precipitation).
- Page 18, Line 340: "We believe that the present study **expands our understanding** of…"

The authors thank the reviewer for the detailed corrections and will apply them thoroughly to the new version of the manuscript.

---

## Author Comment (AC2)

**Response to Anonymous Referee #2**

This study presents an experimental study of the calcite dissolution-induced permeability change in carbonate rocks. The electrical conductivity is used to monitor the experiment, and the measured electrical conductivity is used to infer material pore structure changes. Insights into the link between k and electrical conductivity (and related parameters) are made. In general, this manuscript is well-written and easy to follow. The reported experimental data are new and provide valuable datasets other researchers can use for similar purposes. The data analysis is based on using theoretical models, and thus it gives many insights that otherwise can not be obtained from traditional data analysis.

Therefore, I am in favor of publication.

The authors wish to thank the reviewer for this positive general comment about the manuscript, highlighting the value of the study.

**Specific comments:**

I have several minor comments and hope the authors can consider in revising their manuscript.

1. Different flow conditions

   Page 6 (line 102): the authors mentioned E04 and E05 have different flow conditions. Also, the Peclet number is introduced here to characterize the flow condition. It would be helpful if the authors could talk a bit more about the dissolution pattern related to these two flow conditions. Are we expected to see different flow patterns for these two samples? Based on the experimental results, it seems that the dissolution patterns of the two samples are quite similar.

The point stated by the reviewer is accurate and the authors will add it to the discussion of the results concerning micro CT images and electrical measurements. The dissolution patterns should be quite similar regarding the close values of the Peclet numbers. However, the percolation of E05 happens in almost three times more than the percolation time of E04. Micro CT images show that this difference in flow regime already has an impact on the dissolution pattern. Indeed, even if both samples end up with dominant wormholing, E05 presents a peripheric portion invaded by the acid solution and which has an important impact on the electrical conductivity.

2. Section 3.2

   The authors provide some analysis of the chord length distribution of the two samples before and after the experiment. The current version of the manuscript does not explain very well what a "chord length" is and how "chord length" is related to "pore size" and "pore throat" size. Moreover, the main parameter mj estimated from chord length distribution is also not well explained in terms of its physical meaning. Also, the authors mentioned (Line 230) "the evolution of mj is not a sufficient indicator of the effect of dissolution on the sample". If this is the case, what is the purpose of keeping Section 3.2 then?

First, the authors agree with the reviewer about the need of adding to the manuscript the definition of the chord length distribution. This statistical correlation function has been developed in the study of Torquato and Lu (1993). It is applied to two-phase random media composed of fully penetrable spheres polydisperse in size and corresponds to "the probability of finding a line segment of length $z$ wholly in one of the phases when randomly thrown into the sample." For the study of porous media, the chord length distribution is determined for the phase outside the spheres. From this definition, chord length distribution only has a statistical meaning regarding pore size, but does not provide information on the distribution of pore bodies and pore throats. Second, the parameter $m_j$ is an exponent in capillary size distribution proposed in the study of Jackson et al. (2008), already cited in the manuscript when presenting the analytical model. The authors propose to add the following clarification to the revised version of the manuscript. The proposed capillary size distribution is based on a frequency distribution related to the average capillary radius. The increase of analytical parameter $m_j$ describes how this continuous distribution of pore size is skewed toward smaller capillary radii. Third, the purpose of Section 3.2 is to confront the proposed capillary size distribution, on which the analytical model is based, to the statistical distribution derived from the micro CT images of the real carbonate samples studied in this manuscript. In this section, the capillary size distribution of the analytical model shows accurate agreement with the statistical data. This is a major validation of the hypothesis on which the model is based, which guarantees the reliability of the following interpretation of the model's results. To emphasize that this section is an effective contribution to this study, the preceding lines will be added to the revised manuscript.

Ref: Torquato, S.; Lu, B. Chord-Length Distribution Function for Two-Phase Random Media. Phys. Rev. E **1993**, 47, 2950, doi: 10.1103/PhysRevE.47.2950.

3. I feel that there are too many metrics/parameters discussed in this study. (Yes, one valuable benefit of the experiment in this study is that it can provide different types of data for analysis). A discussion of all of the acquired parameters/properties makes it somehow difficult for readers to understand the key findings of this study. In my opinion, the most interesting and unique finding of this study is that the constrictivity changes significantly during dissolution; in contrast, the hydraulic tortuosity does not change. Other discussions such as k-$\ddot{I}\square$, k-F, and $\Lambda$-k relations, are less important than hydraulic tortuosity and constrictivity. My suggestion is to put more emphasis on the discussion on hydraulic tortuosity and constrictivity.

The authors are grateful to the reviewer for underlying his interest in the significant role played by constrictivity during dissolution. However, the authors do think that depending on the field research of the reader, the other findings are also of interest. To address the reviewer's concerns about prioritization, we propose adding emphasis on constrictivity in the abstract, the title of the dedicated section 3.3, and the conclusions.